# The Time of Day Is Key to Discriminate Cultivars of Sugarcane upon Imagery Data from Unmanned Aerial Vehicle

**Marcelo Rodrigues Barbosa Júnior** [1,*], **Danilo Tedesco** [1], **Vinicius dos Santos Carreira** [1], **Antonio Alves Pinto** [1], **Bruno Rafael de Almeida Moreira** [1], **Luciano Shozo Shiratsuchi** [2], **Cristiano Zerbato** [1] **and Rouverson Pereira da Silva** [1]

1   Department of Engineering and Mathematical Sciences, School of Agricultural and Veterinarian Sciences, São Paulo State University (Unesp), Jaboticabal, São Paulo 14884-900, Brazil; danilo.tedesco@unesp.br (D.T.); vs.carreira@unesp.br (V.d.S.C.); alves.pinto@unesp.br (A.A.P.); b.moreira@unesp.br (B.R.d.A.M.); cristiano.zerbato@unesp.br (C.Z.); rouverson.silva@unesp.br (R.P.d.S.)
2   AgCenter, School of Plant, Environmental and Soil Sciences, Louisiana State University, Baton Rouge, LA 70808, USA; lshiratsuchi@agcenter.lsu.edu
*   Correspondence: marcelo.junior@unesp.br

**Abstract:** Remote sensing can provide useful imagery data to monitor sugarcane in the field, whether for precision management or high-throughput phenotyping (HTP). However, research and technological development into aerial remote sensing for distinguishing cultivars is still at an early stage of development, driving the need for further in-depth investigation. The primary objective of this study was therefore to analyze whether it could be possible to discriminate market-grade cultivars of sugarcane upon imagery data from an unmanned aerial vehicle (UAV). A secondary objective was to analyze whether the time of day could impact the expressiveness of spectral bands and vegetation indices (VIs) in the biophysical modeling. The remote sensing platform acquired high-resolution imagery data, making it possible for discriminating cultivars upon spectral bands and VIs without computational unfeasibility. 12:00 PM especially proved to be the most reliable time of day to perform the flight on the field and model the cultivars upon spectral bands. In contrast, the discrimination upon VIs was not specific to the time of flight. Therefore, this study can provide further information about the division of cultivars of sugarcane merely as a result of processing UAV imagery data. Insights will drive the knowledge necessary to effectively advance the field's prominence in developing low-altitude, remotely sensing sugarcane.

**Keywords:** flight time; NDVI; principal component analysis; reflectance; remote sensing; *Saccharum* spp.; spectral band; UAV; vegetation index

## 1. Introduction

Sugarcane (*Saccharum* spp.) is a specialty cash-crop. It consists of a wealth of benefits to agriculture and bioeconomy, such as producing food, fuel, and feed. Strategically monitoring sugarcane can enable stakeholders in the global sugar-energy system to make precise decisions, streamline workflows, and manage production cost-effectively [1]. Furthermore, it can be an enabler either for HTP or protecting plant breeders' rights (PBRs) of breakthrough genotypes [1]. Remote sensing can be an opening of strategic and catalytic solutions and actions to bring a blueprint for engineering next-generation genotypes into implementation towards a thriving and responsive sector in the near future. However, research and technological development in remote sensing for sugarcane focuses on satellites. System-level studies demonstrate them to effectively acquire useful imagery data only at later stages of development, making the application of them in HTP rather complex and challenging. A disruptive phenotyping intervention for sugarcane is likely to require the acquisition of data either at the plot-level or plant-level; otherwise, it will not function

properly and cost-effectively [2,3]. Therefore, an option to analyze sugarcane phenotypically with greater accuracy and flexibility than what is possible on satellites would be with a UAV.

The importance of UAVs is evident, consisting of plenty of functions. Especially for agriculture, it can be a disruptive solution over precision and digital farming actions, such as realistically monitoring and actively intervening in croplands both autonomously and semi-autonomously at the assistance of an operator [4]. A state-of-the-art crop-sensing UAV can acquire both spatially and temporally high-resolution hyper/multispectral data about the object with pinpoint accuracy in a non-destructive or non-invasive way [5,6]. Even if the target is harder than is accessible to reach through the landscape, such a low-altitude technological platform can enable the user to acquire useful survey-grade data for making decisions at the right time and place in the field [7,8]. Furthermore, its versatility makes it possible to outperform a harsh weather condition, such as cloud and overcast [9,10].

Contemporary literature on remote sensing for agriculture can provide insights into the discrimination of crops upon UAV imagery data. For instance, VIs and discriminant analysis can be a merger for discriminating between varieties of *Olea europaea* [11]. Even if the dataset is structurally complex and heterogeneous, such an approach can summarize it into k-region groups upon VIs. However, the authors [11] stress the importance of introducing into the biophysical model an additional level of relevant information about spatiotemporal and environmental conditions to bring it into a higher generalist level of abstraction. Most importantly, they point out the need for previously defining groups to calculate discriminants; thus, handling a more suitable multivariate statistical approach such as principal component analysis (PCA) may be necessary. The PCA can calculate best discriminant latent hits without foreknowledge about groups. By reviewing the study by Caruso et al. [12] and also discriminating between varieties of European olive, the normalized difference vegetation index (NDVI) and fruit yield can be useful information for the discrimination of experimental data and making early-stage decisions in the field for successful harvesting. Furthermore, the study by Galidaki, Panagiotopoulou, and Vardoulaki [13] on the deployment of a UAV into the precision vitiviniculture can provide a timely reference on the technical viability of discriminating cultivars of Vitis sp. upon multispectral imagery data. The discrimination could be satisfactorily accurate at pixel-level and even plot-level, which is not always possible in orbital remote sensing. Therefore, research into UAV's imagery data for discriminating between crops concentrates on orchards. In contrast, systematic studies and analyzes on sugarcane often are not available from typical literature, driving the need for further in-depth investigation.

To the best of our knowledge, no in-depth investigation exists on the feasibility of operating an UAV to acquire imagery data and discriminate between cultivars over sugarcane-producing areas, both experimentally and commercially; nor have there been systematic analyses of flight features that are still sufficient and conducive to perform seamless remote sensing. The integrative review by Barbosa Junior et al. [4] states the importance of the UAV as a disruptive solution to reshape the monitoring and management of sugarcane over readily accessible crop-sensing, crop-spraying, and stuff-releasing models. The authors [4] provide systematic and meta-analytical insights into the operation of such multi-objective devices at experimental and commercial scales. However, they identify inconsistent and insufficient description for the flight's features across studies. Full-text papers often do not describe the mission clearly and completely. For instance, duration of flight and time of day are missing from the methodological sections of papers. The time of day can especially negatively impact the remote sensing by introducing radiometric inconsistencies into the surveying, from taking-off to landing [4]. The choice of time of day is key to perform the flight on field. A suitable setup can enhance the monitoring of spectral variability, whether for fertilizing in cotton [14], mulching on tomato [15], and phenotyping in maize [16], without environmental noise and degradation of the image's quality.

Reflectance is specific to crops. However, it can widely fluctuate throughout the day, making it challenging and potentially biasing the acquisition of spectral data to compare

multiple phenotypes and even analyze the growth, development, and vigor of agricultural commodities [14–16]. Since UAV is versatile and can outperform a harsh environment, it can drive the need of overcoming limitations of time of day to remote sensing, especially for sugarcane. The primary objective of this study was therefore to analyze whether it could be possible to discriminate market-grade cultivars of sugarcane upon UAV imagery data. A secondary objective was to analyze whether the time of day could impact the expressiveness of spectral bands and VIs in biophysical modeling.

## 2. Material and Methods

### 2.1. Study Area

The field study was conducted in an experimental area located at São Paulo State University, Jaboticabal, Brazil (Figure 1). The region has low slope (0–8%) and the soil is Oxisol [17]. The region's climate is characterized as tropical Aw [18], with a dry winter season; the mean annual precipitation and temperature are 1460 mm and 22.6 °C, respectively. Five sugarcane cultivars were planted in January 2021, and spectral information was collected at the tillering stage (90 days after planting the pre-sprouted seedlings). A total of 300 plots were used in this study, each plot corresponding to one row of sugarcane. The field was divided into 1.5 m × 10 m plots to capture spectral bands and VIs over a single day. The experimental field was divided into five sections based on the sugarcane cultivars, namely RB966928, RB867515, IACSP95-5094, CTC9001, and CTC-4. Both sections were composed of 60 plots, where the spectral attributes were extracted.

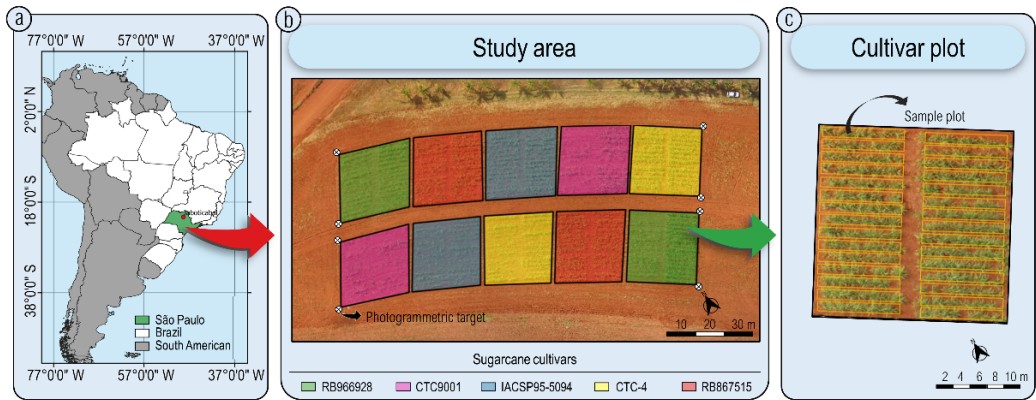

**Figure 1.** (**a**) Location of the field area in Brazil. (**b**) UAV orthomosaic of the study area highlighting cultivars. (**c**) One cultivar from the field with overlapping polygons on each sugarcane row.

### 2.2. Data-Acquisition Platform's Description

A multi-rotor UAV (Tarot, Ironman 650, Wenzhou, Zhejiang, China) was equipped with a multispectral sensor (MicaSense RedEdge-M, MicaSense Inc., Seattle, WA, USA) for collection of spectral data. The captured bands by the sensor were Blue, Green, Red, RedEdge, and NIR (wavelengths are presented in Table 1) with an image size of 1280 × 960 pixels. The UAV was controlled by flight planning software (Mission Planner, ArduPilot, Canberra, Australia) and a mission planner PC, used to automate parallel axis flight lines and image capture across the field. To guarantee the same geographical position of the five orthomosaic, we deployed eight photogrammetric targets in the field and then performed the positional setting using Georeferencer plugin from the QGIS software (Free software Inc, Boston, MA, USA). UAV data were collected in the 2021 growing season. Five flights were performed throughout the day (Table 2). In this study, the images from each flight were radiometrically calibrated using radiance values from a calibrated reflectance panel (MicaSense Inc, Seattle, WA, USA), recorded from the camera before each flight. Radiance, temperature, and humidity also were collected automatically by a weather station near experimental area to further characterize the environment (Figure 2).

**Table 1.** Wavelength and full width at half maximum (FWHM) for bands present in Micasense Red-Edge-M sensor.

| Band Number | Band Region | Center Wavelength (nm) | Bandwidth FWHM (nm) |
|:---:|:---:|:---:|:---:|
| 1 | Blue | 475 | 20 |
| 2 | Green | 560 | 20 |
| 3 | Red | 668 | 10 |
| 4 | RedEdge | 717 | 10 |
| 5 | NIR | 840 | 40 |

**Table 2.** UAV data collection timeline and sensor wise flight specification.

| Flight Time | | Flight Altitude (m) | Number of Images | Overlap (%) | | GSD (cm) | Sensor Inclination (°) |
|:---:|:---:|:---:|:---:|:---:|:---:|:---:|:---:|
| Start | End | | | Side | Front | | |
| 8:01 AM | 8:12 AM | 30 | 1955 | 80 | 70 | 2.25 | 90 |
| 10:03 AM | 10:13 AM | 30 | 1960 | 80 | 70 | 2.15 | 90 |
| 12:00 PM | 12:11 PM | 30 | 1965 | 80 | 70 | 2.13 | 90 |
| 2:02 PM | 2:12 PM | 30 | 1965 | 80 | 70 | 2.06 | 90 |
| 4:00 PM | 4:10 PM | 30 | 1950 | 80 | 70 | 2.14 | 90 |

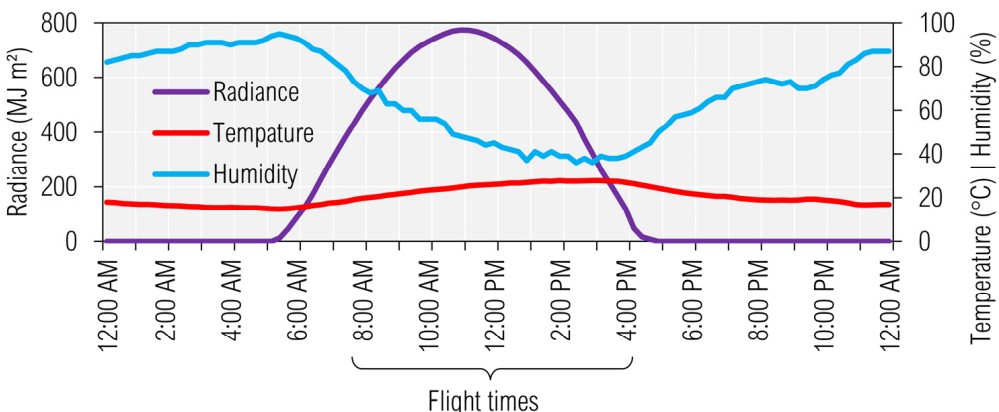

**Figure 2.** Radiance, temperature and humidity data collected at 15-min intervals on 21 April 2021.

### 2.3. Structure from Motion Photogrammetric Processing

The multispectral sensor collected data throughout one day in the experimental field. The multispectral images were processed using SfM software (Agisoft Metashape Professional 1.5.5, Agisoft, St. Petersburg, Russian) to stitch the images. SfM photogrammetry is a photogrammetric range imaging technique for estimating three-dimensional structures from a set of projected two-dimensional images [19]. In this process, homologous information from overlapping images is identified by feature matching algorithms so that clustering of the images occurs. After that, a point cloud is densified, and finally this dense point cloud can be used to generate a digital elevation model (DEM) and an orthomosaic model [20].

### 2.4. Image Segmentation

Orthomosaics were segmented to remove the interference of soil background. We created a training dataset with two classes, each class representing plant and soil background, respectively. We considered all cultivars in the study field as a single class. Thus, in each image acquisition time, 2048 and 240 plant and soil classes were created, respectively. In addition, to allow for independent validation, a data set was created with 512 plants and 60 soil samples. In short, 80% of the data was used for training and 20% for validation. Here, we used the Random Forest classifier, as per recommendation [21–23]. The hyperparameters of the classifier were optimized based on the input information (GridSearchCV)

and 10-fold cross-validation was performed to avoid overfitting. The whole procedure was performed with tools from the Scikit-learn package [24]. Finally, a non-binary mask layer was established on each classified orthomosaic in order to filter out only plants. The segmentation applied had an accuracy higher than 97% for all flights.

*2.5. Spectral Data Extraction*

Spectral band information was obtained from the sample plots of the study field. For this step, we used the Mask and Zonal Statistical tools from the QGIS. In addition to the spectral bands described in Table 1, we also used VIs to differentiate sugarcane cultivars. VIs consisted of spectral bands and were developed to evaluate spectral behavior of plants and are widely used in numerous agricultural applications. Here, we chose VIs based on visible and non-visible wavelengths (Table 3).

**Table 3.** Vegetation indices used in this study.

| VI | Nomenclature | Equation |
|---|---|---|
| NDVI | Normalized Difference Vegetation Index | $\frac{NIR - Red}{NIR + Red}$ |
| NDRE | Normalized Difference Red Edge Index | $\frac{NIR - RedEdge}{NIR + RedEdge}$ |
| EVI | Enhanced Vegetation Index | $\frac{2.5 \cdot (NIR - Red)}{NIR + 6.0 \cdot Red - 7.5 \cdot Blue + 1}$ |
| VARI | Visible Atmospherically Resistant Index | $\frac{Green - Red}{Green + Red - Blue}$ |
| GLI | Green Leaf Index | $\frac{2 \cdot Green - Red - Blue}{2 \cdot Green + Red - Blue}$ |

*2.6. Data Analysis*

Principal component analysis (PCA) was applied as a multivariate statistical approach to transform the spectral information into a set of orthogonal uncorrelated variables (principal components—PCs). Here, our objective was to explore the possibility of grouping sugarcane cultivars with similar spectral profiles. For this, two datasets were obtained. The first one consisted of the means of spectral bands (Table 1) for each cultivar. The second was formed by the means of VIs (Table 3) for each cultivar.

## 3. Results

*3.1. Spectral Variation throughout the Day*

The primary hypothesis of this study was that UAV will act as a low-altitude remote sensing platform and acquire useful imagery data to discriminate cultivars upon spectral bands and VIs. By analyzing the reflectance of cultivars (Figure 3), it should be stated that hypothesis was indeed validated. The action of flying the UAV on the experimental field to acquire imagery data every 2 h, from 8:00 AM to 4:00 PM, adequately captured the temporal variability in reflectance throughout the day, irrespective of cultivar. In contrast, cultivars likely were not spectrally distinctive in space, where plot-level zones with higher reflectance did relate to longer wavelengths detectable from the multispectral optical sensors onboard the UAV. The choice of time of day to perform the flight impacted the reflectance, especially for longer wavelengths. The NIR provided the largest variation in reflectance. In contrast, B provided the smallest variation in reflectance. Furthermore, cultivars reflected maximum proportion of perpendicularly incident sunlight through the R at 12:00 PM, which corresponded to the time of the highest temperature and radiance (Figure 2).

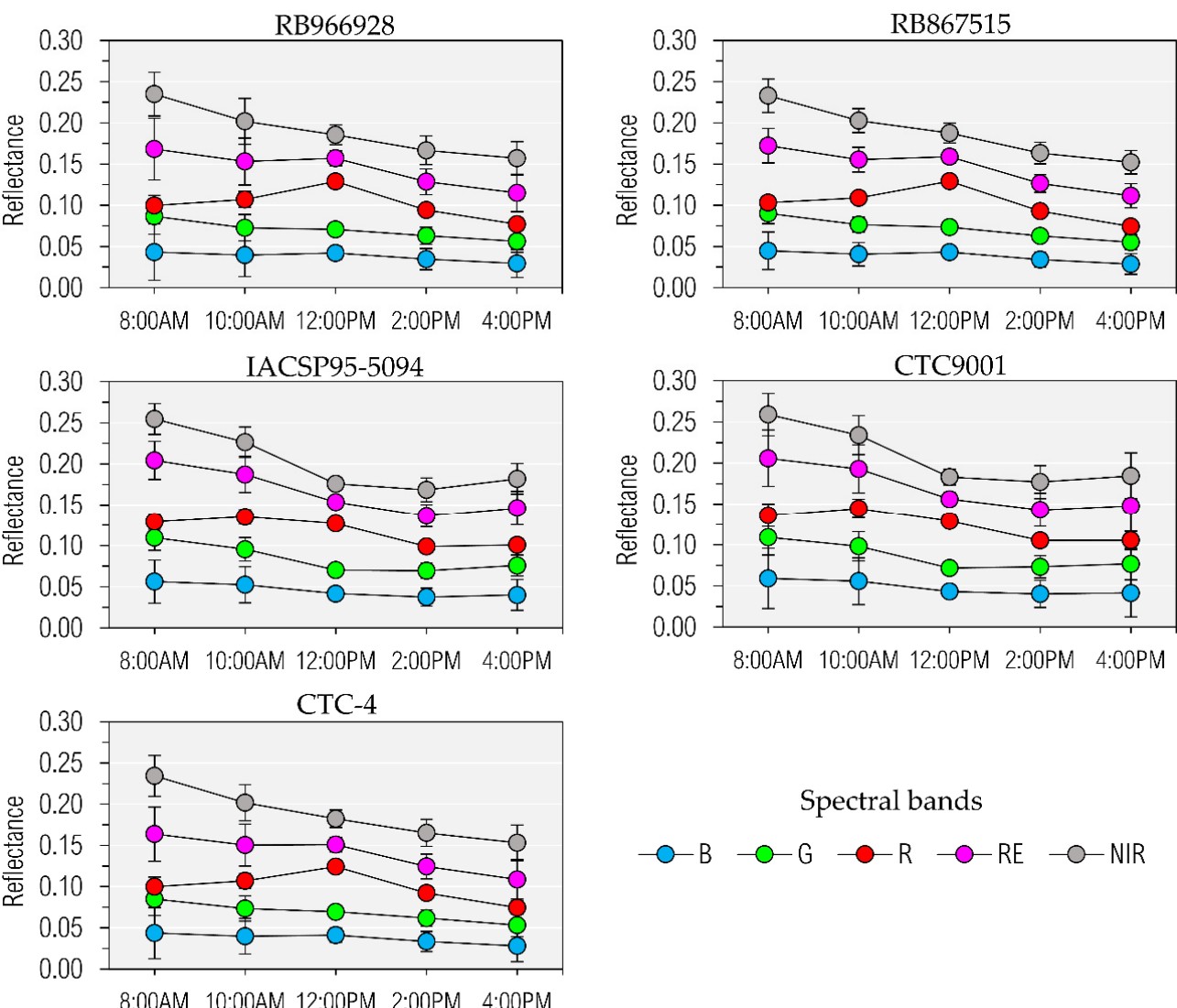

**Figure 3.** Spectral behavior of sugarcane cultivars at five times of day. The diagram displays dots consisting of 60 plots (mean ± standard deviation).

*3.2. Discriminating on PCA for Sugarcane Cultivars upon Spectral Bands and Vegetation Indices*

The secondary hypothesis of this study was that the time of day will determine the expressiveness of visible (R, B, G) and non-visible (RE and NIR) spectral bands and VIs (NDVI, NDRE, EVI, VARI, and GLI) in the discrimination of cultivars upon UAV imagery data. By analyzing the factorial map (Figure 4), it should be given the statement of discriminant eigenvalue. The PCA adequately summarized the high-resolution raw imagery data into the orthogonal subsets, namely Dim 1 and Dim 2. Such latent hits together explained about 90% of the total variability in reflectance. A cumulative percentage of variance equal to 80% or greater is sufficient to address a reliable multivariate statistical analysis.

Spectral bands had significant positive correlation with another one. However, exceptionality for 12:00 did not show a linear relationship between NIR and R. While the NIR contributed most effectively to the characterization of the cultivar RB966928, visible bands contributed most effectively to the characterization of the cultivar CTC9001; especially the R. This supported particular spectral profiles and, most notably, the possibility of discriminating on PCA for spectrally distant cultivars upon long-wavelength spectral bands. In contrast, CTC-4, IACSP95-5094, and RB867515 moved toward the quadrants consisting of low loadings with spectral bands, irrespective of time of day. Generally, NIR

and R provided the best insights into the discrimination of cultivars, making them the most reliable spectral bands to scale, especially for RB966928 and CTC9001.

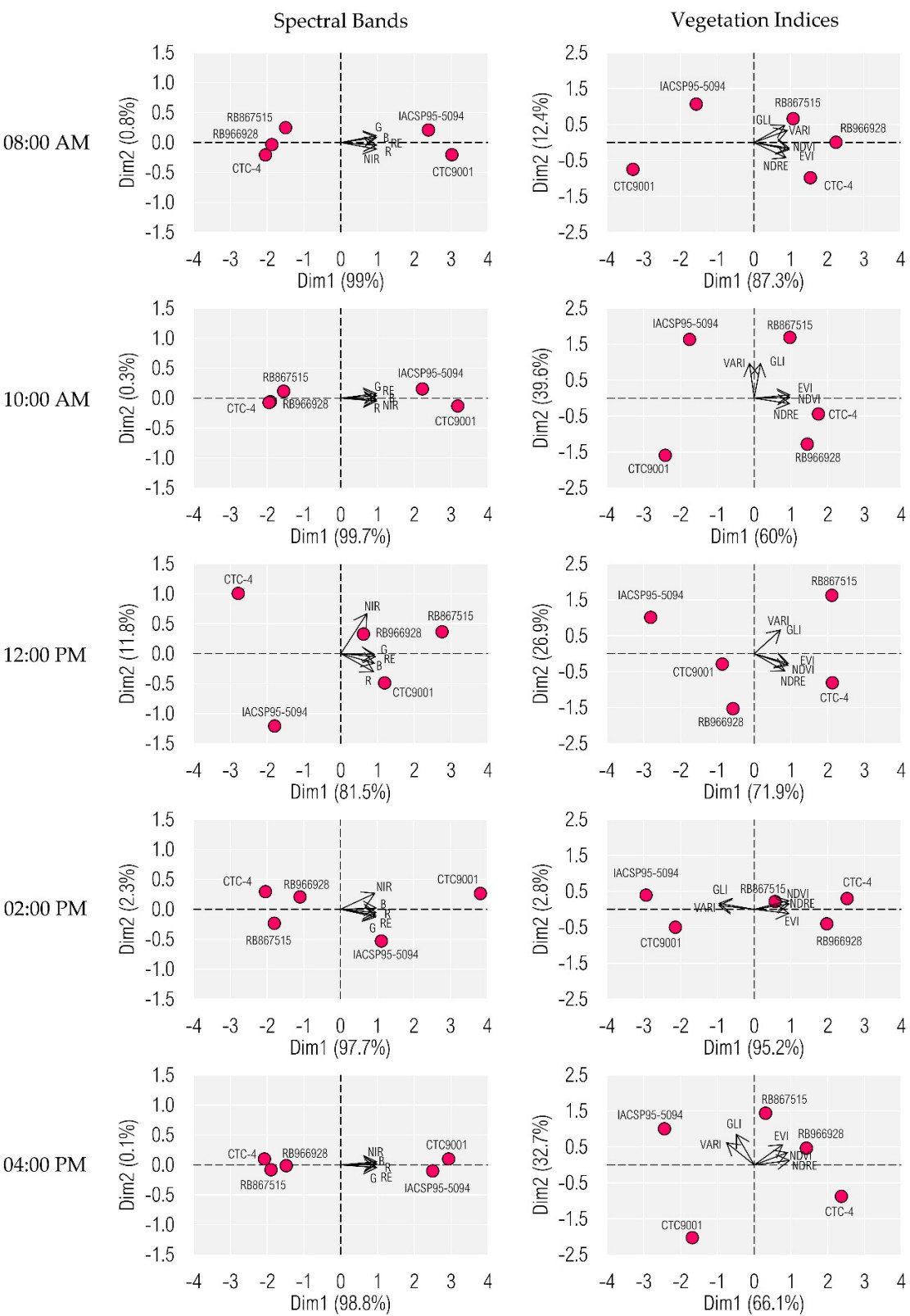

**Figure 4.** Principal component analysis based on the mean Euclidean distance between sugarcane cultivars for different spectral bands and vegetation indices.

By analyzing the mapping of cultivars to VIs, NDVI, NDRE, and EVI moved forward in the space of Dim 1, making them eigenvectors farther from the graphic origin and opposite to VARI and GLI. Such mathematical transformations of visible spectral bands move backward in the space of Dim 1, making them "close-to-the-origin" eigenvectors, especially for 2:00 PM and 4:00 PM. Visible VIs had significant positive correlations another one, irrespective of time of day. In contrast, these had no linear relationships with VIs consisting of invisible spectral bands for all times but 2:00 PM. Plot-level imagery data for the specific flight at 2:00 PM produced negative correlations between visible and invisible VIs. While the R did not contribute to discriminating IACSP95-5094, its integration to G and B to yield VARI and GLI allowed for discriminating such cultivar and also RB867515. These visible VIs had significant correlations with both IACSP95-5094 and RB867515, making it possible to separate them graphically with greater adequacy than what was possible on NDVI and NDRE. Such invisible VIs had a positive correlation with another one through the NIR, and effectively separated the cultivars; especially CTC-4.

## 4. Discussion

### 4.1. Spectral Variation throughout the Day

The multispectral sensor enabled the UAV to capture the spectral variation throughout the day. Cultivars reflected the maximum portion of perpendicularly incident sunlight on the canopy through the NIR (Figure 3). Therefore, they most effectively absorbed the energy in the photosynthetically active radiation (PAR) of the spectrum between 400 nm and 700 nm, reflecting longer wavelengths [25]. Furthermore, NIR produced the largest variation in reflectance throughout the day, especially for flights at 8:00 AM and 4:00 PM. The angle of incident sunlight and its environmental fluctuation can impact the NIR-specific reflectance of crops, such as wheat [26], barley [27] e rapeseed [28], and sugarcane in this study. Other limiting factors may include leaf's components and photosynthetic processes. Therefore, further in-depth investigation must concentrate on analyzing how morpho physiology of cultivars and the environment interact to determine the reflectance.

Spectral variations are indicators for the impact of environmental factors, such as radiance, temperature and relative humidity on crops [29]. For instance, potato and sugar beet [30] and rice [31] are highly responsive to environmental fluctuations. They are likely to have heterogeneous peaks of photosynthetic activity and reflectance throughout the day, supporting trends in this study for the spectral behavior of cultivars of sugarcane. While the NIR-specific reflectance was the highest at 8:00 AM, cultivars reflected the most sunlight through the R at 12:00 PM. Such long-wavelength bands are highly sensitive to synthesis and absorptive activity of chlorophylls [32,33]. The time of day can either upregulate or downregulate the emission of fluorescence from chlorophylls, changing the reflectance [34–36]. Hoel and Solhaug [37] provided a timely reference on the relationship between irradiance and activity of chlorophylls. The higher the irradiance the lower the activity of such light-harvesting pigments, supporting lower values of NDVI at 12:00 PM in this study. At 12:00 PM, the irradiance was the highest, and thus the VIs decreased. The decrease of temperature and irradiance at earlier (8:00–10:00 AM) and (2:00–4:00 PM) times of day enabled the cultivars to reflect more of energy in the PAR detectable from the multispectral sensor onboard of UAV, and thus the VIs increased. Earlier studies on soybean [38] and cotton [14] reported similar trends to those in this study, although such crops and sugarcane could be spectrally distinctive.

### 4.2. Discrimination of Cultivars upon Spectral Bands and VIs

Once we have not grouped the cultivars from the premise dataset (RB966928, RB867515, IACSP95-5094, CTC9001, and CTC-4), we hypothesized that PCA will calculate best discriminants from predictive datasets either consisting of spectral bands (R, B, G, RE, and NIR) or VIs (NDVI, NDRE, EVI, VARI, and GLI) to separate them graphically. By analyzing the outcome of PCA-evolved discrimination, the technique should be validated as a reliable multivariate approach to deal with separating cultivars according to similar or dissimilar

spectral features without the need of foreknowledge about groups. This could overcome the limitations of discriminant analysis, which can calculate best discriminants only if groups are defined by the user [11].

Spectral bands and VIs proved to be useful discriminants. However, existing redundancy in the factorial map could make it hard to interpret the outcome. Most importantly, it could bias the algorithm, so it may not accurately map cultivars to imagery data as we might expect on data without misinformation-to-information overlapping. An option to optimize the discrimination would be redundancy analysis (RDA). The PCA and RDA work analogously [39]. Both summarize high-dimensionality data into something meaningful and easier to analyze. However, PCA is not constrained and likely to search for any variable that best explains the variation, whereas RDA is a constrained version of PCA and will search for best explanatory variables. Therefore, further research should focus on analyzing whether RDA could calculate discriminants better than PCA, especially for cultivars (e.g., CTC-4 and RB867515) that are weakly correlated with spectral information (e.g., G and B bands). If our perspective is right, constraining the analysis will improve modeling spectral variation throughout the day.

Especially the NDVI has proven accurate across studies on identifying varieties/cultivars of sugarcane [40], wheat genotypes [41] and olive cultivars [11]. However, NDVI is highly responsive to soil, driving the need of segmenting the image, such as the photogrammetric processing in this study. The removal of soil out of the image can enable the discrimination of potato upon the NDVI; otherwise, it becomes rather complex and not accurate. An option to discriminate crops based on raw imagery data would be EVI. Such VI is not sensitive to soil brightness and can perform better NDVI, especially for dense vegetation [42]. Another limiting factor to NDVI refers to environmental fluctuations through the day, such as in this study and in the investigation on predictive imagery data for wheat in early development stage [41]. The NDVI decreases at 11:00 AM and 1:00 PM, as irradiance and temperature increase and stress the crop [43].

Therefore, by considering the research into modeling crops upon the NDVI, this study provides alternative VIs and even spectral bands to overcome limitations towards discriminating cultivars with greater accuracy. However, time-dependent similar spectral features could make it challenging to map them. Spectral properties depend on a wide range of factors, such as physiological condition and surrounding sources of stress both biotically and abiotically [36]. An accurate intraspecific identification of crops—for instance, banana [44] and rice [23]—at early stages of development is not always possible with conventional techniques. Even employing state-of-the-art methods such as remote sensing, computational processing of image and explanatory data analytics in this study may not be sufficient to discriminate cultivars of sugarcane which can be spectrally similar at tillering. Therefore, another direction to future research is to analyze whether an UAV is capable of acquiring useful imagery data throughout the phenological stage of differentiation to develop a thriving and responsive decision-supporting system.

*4.3. The Value of this Study to Advance the Field of UAVs for Sugarcane and the Ways Forward*

The flight time is mainly related to field size, equipment availability, and accuracy requirements. In UAV applications for obtaining spectral information from plants, setting the correct flight time is necessary to obtain the most accurate data and can solve substantial problems regarding weather variations. Currently, weather variations provide considerable challenges for sensor-based technology applications, which can often render these applications "precisely inaccurate" [45]. Since research into low-altitude remote sensing for sugarcane is progressing [4], this timely study can demonstrate and validate the applicability of operating an UAV to acquire useful imagery data to discriminate cultivars, whether for precision management or HTP. The time of day is key to remotely sense the reflectance and conducive to separating spectrally similar or dissimilar cultivars through PCA. Visible (R, B, G) and non-visible (RE and NIR) spectral bands and VIs (NDVI, NDRE, EVI, VARI, and GLI) can prove to be discriminant eigenvectors to address an accurate

biophysical modeling. They can enable the need of defining the time of day to perform the flight on field to extract particular spectral information about cultivars with plot-level accuracy and flexibility, what is not always possible on satellites.

By analyzing the operationality, flights in this study lasted for 10 min. This duration was sufficient to survey the experimental field without a critical spectral variation between the beginning and the end of every mission. However, large fields could be conducive to a high temporal heterogeneity in reflectance during the flight. Therefore, the methodology in this study should be given an adaptation to address consistent remote sensing at a commercial scale, from take-off to landing. For instance, stakeholders (e.g., companies, producers and assistants) should set up the UAV to fly for up to 2 h. They should perform the flight at higher altitudes to work in more areas with less time. However, they should balance between operationality and quality of imagery data (resolution); otherwise, they may not bring a cost-effective project into implementation. Furthermore, they should split the field into homogeneous sections and make analyses and decisions within them and not as a whole. These are general recommendations, however. User demand, agribusiness model, and technological level should be considered in scaling the concept of this study, which is still at an early stage of development. Further research into techno-economic analysis and life-cycle assessment is needed to take off from the academic space in order to position it into the real world.

## 5. Conclusions

Unmanned aerial vehicles can acquire useful imagery data to discriminate cultivars of sugarcane upon spectral bands and VIs. R and NIR are the best discriminant spectral bands for cultivars at 12:00 PM. Irrespective of time of day, VARI, GLI, NDVI, and NDRE are reliable VIs to scale. Analytical insights into the conceptual and technical ramifications of this study are timely. They will likely provide further knowledge about discrimination of cultivars upon a UAV's imagery data to effectively advance the field's prominence in developing low-altitude remote sensing for sugarcane, whether for precision management or HTP. Therefore, stakeholders will benefit from them intervening in the field at the right place and time with greater accuracy (plot-level or plant-level) and flexibility than what is possible with satellites.

**Author Contributions:** Conceptualization, M.R.B.J.; methodology, M.R.B.J. and D.T.; validation, M.R.B.J. and D.T.; formal analysis, M.R.B.J.; investigation, M.R.B.J.; data curation, M.R.B.J. and D.T.; writing—original draft preparation, M.R.B.J., V.d.S.C. and A.A.P.; writing—review and editing, M.R.B.J., D.T., V.d.S.C., A.A.P., B.R.d.A.M., L.S.S., C.Z. and R.P.d.S.; visualization, M.R.B.J., D.T., V.d.S.C., A.A.P., B.R.d.A.M., L.S.S., C.Z. and R.P.d.S.; supervision, R.P.d.S.; project administration, R.P.d.S.. All authors have read and agreed to the published version of the manuscript.

**Funding:** We acknowledge support for the publication of this work by the Publishing Fund of Graduate Program in Agronomy (Soil Sciences).

**Acknowledgments:** We would like to acknowledge the Coordination for the Improvement of Higher Education Personnel (Capes), for the scholarship (code 001) to the first author, and the Laboratory of Machinery and Agricultural Mechanization (LAMMA) of the Department of Engineering and Mathematical Sciences for the infrastructural support.

**Conflicts of Interest:** The authors declare no conflict of interest.

## Abbreviations

| | |
|---|---|
| DEM | digital model elevation |
| EVI | enhanced vegetation index |
| GLI | green leaf index |
| HTP | high-throughput phenotyping |
| NDRE | normalized difference rededge index |
| NDVI | normalized difference vegetation index |
| PAR | photosynthetically active radiation |
| PCA | principal components analysis |
| PBRs | protecting plant breeders |
| RDA | redundancy analysis |
| SfM | structure from motion |
| UAVs | unmanned aerial vehicles |
| VARI | visible atmospherically resistant index |
| VIs | vegetation indices |

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
