# Peer review of "The Time of Day Is Key to Discriminate Cultivars of Sugarcane upon Imagery Data from Unmanned Aerial Vehicle"

_drones, doi:10.3390/drones6050112_

Round 1

Reviewer 1 Report

Thanks for the opportunity to review this paper.

In my opinion it is a very interesting work, because  presents a study for the discrimination of sugar cane cultivars through the determination of spectral indices from images captured by unmanned aerial vehicles captured at different times of the day. However, the manuscript is not yet ready for publication, but has much promise if the authors are willing to undertake some revisions.

Lines #103-106. Figure 1 has an error, the “b” is not indicated in the explanatory text. Looking at the figure it seems to be deformed (please check). Plot c is disoriented, it does not correspond to any of the 19 plots that appear in the figure. A scale should be put, even if it is graphic, in figures b and c.

Line #123 Point: 2.3.- 2.3. Structure from Motion photogrammetric processing. In the workflow there is no mention of photogrammetric orientation: no photocontrol points have been placed to guarantee the correct georeferencing of the drone flights to allow comparison between the orthophotos obtained at different times of the day. This would be necessary in order to be able to make these comparisons.

Line # 107. Item 2.2. Sensors and aerial platfor. The capture of information for the radiometric calibration of the camera is not explained in the text: calibration panels, white/black bodies needed to determine the reflectivity and to make atmospheric corrections. This aspect is critical for a study of this type.

Line #179-180 Figure 3. It is not very explanatory, if what is intended is to differentiate between cultivars it would be clearer to represent a graph for each band where the cultivars are represented, this would define the possibility of spectral separation of the cultivars. Please correct.

Reviewer 2 Report

Reviewer’s comments on manuscript titled “UAV imagery can differentiate sugarcane cultivars and the flight time is a key to a reliable approach”.

GENERAL COMMENTS

The authors have presented a study evaluating the use of UAV-based imagery data acquired using a multispectral sensor for sugarcane cultivar differentiation. This is a straightforward study and lacks novelty from the sensing part. There have been numerous studies which have utilized spectral data for differentiation of crop cultivar using ground based or remote sensing. UAV is just a platform to carry the sensor, and the authors could have explored the UAV flight attributes (flight speed, height etc.) to see how such parameters could be optimized for a better classification or differentiation with a higher accuracy. Nonetheless, the manuscript may be of interest to plant breeders as use of UAVs in agriculture is gaining momentum. The manuscript is written well, however, I have a few concerns which need to be addressed to make this manuscript acceptable for publication. For more detailed comments, please see the attached file.

General comment 1: It has been already established extensively that the best time to acquire UAV imagery is around noon time. Therefore, it is beyond comprehension why the authors chose to look at different time of the day as a variable in this study. Ultimately, they concluded the same thing from their study.

General comment 2: The title should be revised as flight time may be interpreted as the actual duration the UAV flies for. Instead, the authors may use “time of the day”.

General comment 3: The introduction section needs to be improved by adding more relevant/recent studies. The problem statement doesn’t justify the objectives, and needs to be revisited. The authors justify the objectives by talking about chlorophyll changes in the plants as impacted by irradiance and other environmental conditions. If this is true, a simple sensor can do this work with less resources and logistics. The use of UAVs can generate a high resolution data in a relatively quick time, and I believe the authors should approach their objectives from the perspective of the advantages of UAVs.

General comment 4: Some key information is missing in the material and method section, like flight height, sensor inclination etc. Please see the attached file for specific comments in this section.   

General comment 5: The results section is very small, and I would suggest merging it with the discussion section.  The discussion section contains some information which has already been mentioned in the introduction section (Line 214-226).

General comment 6: The authors used PCA to see the separation among the cultivars. It’s unsure why they used this method when several other methods are available.

Author Response

The reviewer commented "please see the attached file". But, it was not found from the online platform. Therefore, we revised the text based on comments and suggestions below.

Reviewer 3 Report

The topic is intersting and well organized. Here you can find my comments:

INTRODUCTION: Please, consider also the use of remote sensing techniques in the monitoring of environmental and ecohydraulic phenomena (i.e.,

McGlade, J.; Wallace, L.; Reinke, K.; Jones, S. The Potential of Low-Cost 3D Imaging Technologies for Forestry Applications: Setting a Research Agenda for Low-Cost Remote Sensing Inventory Tasks. Forests 2022, 13, 204. https://doi.org/10.3390/f13020204.

Guimarães, N.; Pádua, L.; Marques, P.; Silva, N.; Peres, E.; Sousa, J.J. Forestry Remote Sensing from Unmanned Aerial Vehicles: A Review Focusing on the Data, Processing and Potentialities. Remote Sens. 2020, 12, 1046. https://doi.org/10.3390/rs12061046

Lama, G.F.C.; Sadeghifar, T.; Azad, M.T.; Sihag, P.; Kisi, O. On the Indirect Estimation of Wind Wave Heights over the Southern Coasts of Caspian Sea: A Comparative Analysis. Water 2022 4, 843. https://doi.org/10.3390/w1406084.

Gale, M. G., Cary, G. J., Van Dijk, A. I., & Yebra, M. (2021). Forest fire fuel through the lens of remote sensing: Review of approaches, challenges and future directions in the remote sensing of biotic determinants of fire behaviour. Remote Sensing of Environment, 255, 112282.

He, J., & Barton, I. (2021). Hyperspectral remote sensing for detecting geotechnical problems at Ray mine. Engineering Geology, 292, 106261.

Godone, D.; Allasia, P.; Borrelli, L.; Gullà, G. UAV and Structure from Motion Approach to Monitor the Maierato Landslide Evolution. Remote Sens. 2020, 12, 1039. https://doi.org/10.3390/rs12061039.

METHODOLOGY: Please add some figures showing the spefic UAV and sensors employes in your study case. This is extremely important for the clarity of the entire section.

Please report a figure for each sub-section. The reader must be guided in the understanding of the scientific message proposed by the Authors.

I will reconsider the article after major revision.

Reviewer 4 Report

In this paper the authors analyze the sensitivity of spectral bands and vegetation indices (VIs) to differentiate sugarcane cultivars. Five flights throughout the day were performed in unmanned aerial vehicles (UAVs) their results showed that cultivars were separated and that the time of flight clearly influenced these results. For spectral bands, the best results for differentiating cultivars were at 12:00 PM and for VIs did not a specific time.

This study very interesting because provides a new approach for differentiation of sugarcane cultivars at field scale, and demonstrate the potential application of UAV on sugarcane crop, enabling prospective stakeholders to make decisions with higher confidence.

The manuscript is clear, relevant and presented in a well-structured manner.

There are some comments

Page 1, line 7 the principal author email is missing.

Page 1, line 20 “12:00 AM” should be “12:00 PM”

Page 2, line 62 “UAV’s” should be “UAVs”

Page 3, line 104 “UAV orthomosaic of the study area highlighting cultivars.” should be “(b) UAV orthomosaic of the study area highlighting cultivars.”

Page 3, line 1011 “were B, G, R, RE, and NIR” should be “were Blue, Green, Red, RedEdge, and NIR”

Page 4, line 132 “a DEM” should be “a digital elevation model (DEM)”

In table 3, page 5. In the case of equations, if you are using Word, please use either the Microsoft Equation Editor or the MathType add-on.

Page 10, line 323 “M.R.B.J., V.d.S.C.,” should be “V.d.S.C.,”

Round 2

Reviewer 1 Report

The authors have made the corrections recommended at the revision stage. In my opinion the paper is ready for publication. 

Reviewer 3 Report

The authors considered in the first round only one of the reviewer's indications.

I suggest correcting also the other two issues for the next round.

INTRODUCTION: Please, consider also the use of remote sensing techniques in the monitoring of environmental and ecohydraulic phenomena (i.e.,

McGlade, J.; Wallace, L.; Reinke, K.; Jones, S. The Potential of Low-Cost 3D Imaging Technologies for Forestry Applications: Setting a Research Agenda for Low-Cost Remote Sensing Inventory Tasks. Forests 2022, 13, 204. https://doi.org/10.3390/f13020204.

Guimarães, N.; Pádua, L.; Marques, P.; Silva, N.; Peres, E.; Sousa, J.J. Forestry Remote Sensing from Unmanned Aerial Vehicles: A Review Focusing on the Data, Processing and Potentialities. Remote Sens. 2020, 12, 1046. https://doi.org/10.3390/rs12061046

Lama, G.F.C.; Sadeghifar, T.; Azad, M.T.; Sihag, P.; Kisi, O. On the Indirect Estimation of Wind Wave Heights over the Southern Coasts of Caspian Sea: A Comparative Analysis. Water 2022 4, 843. https://doi.org/10.3390/w1406084.

Gale, M. G., Cary, G. J., Van Dijk, A. I., & Yebra, M. (2021). Forest fire fuel through the lens of remote sensing: Review of approaches, challenges and future directions in the remote sensing of biotic determinants of fire behaviour. Remote Sensing of Environment, 255, 112282.

He, J., & Barton, I. (2021). Hyperspectral remote sensing for detecting geotechnical problems at Ray mine. Engineering Geology, 292, 106261.

Godone, D.; Allasia, P.; Borrelli, L.; Gullà, G. UAV and Structure from Motion Approach to Monitor the Maierato Landslide Evolution. Remote Sens. 2020, 12, 1039. https://doi.org/10.3390/rs12061039.

METHODOLOGY: Please add some figures showing the spefic UAV and sensors employes in your study case. This is extremely important for the clarity of the entire section.

I will reconsider the article after major revisions.